# Vertebral Augmentation: Is It Time to Get Past the Pain? A Consensus Statement from the Sardinia Spine and Stroke Congress

**DOI:** 10.3390/medicina58101431

**Published:** 2022-10-11

**Authors:** Joshua A. Hirsch, Chiara Zini, Giovanni Carlo Anselmetti, Francisco Ardura, Douglas Beall, Matteo Bellini, Allan Brook, Alessandro Cianfoni, Olivier Clerk-Lamalice, Bassem Georgy, Gianluca Maestretti, Luigi Manfré, Mario Muto, Orlando Ortiz, Luca Saba, Alexis Kelekis, Dimitrios K. Filippiadis, Stefano Marcia, Salvatore Masala

**Affiliations:** 1Department of Radiology Massachusetts General Hospital, Harvard Medical School Boston, Boston, MA 02114, USA; 2UOC Radiologia Firenze 1, USL Toscana Centro, 50100 Firenze, Italy; 3Interventional Radiology Unit, European Institute of Oncology IEO Milan, 10060 Milan, Italy; 4Spine Unit, Orthopedics and Traumatology Department, University Clinical Hospital of Valladolid, 47005 Valladolid, Spain; 5Comprehensive Specialty Care, Oklahoma City, OK 7301, USA; 6UOC Neuroimmagini, Neuroradiologia Clinica e Funzionale Dipartimento di Scienze Neurologiche e Motorie Azienda Ospedaliera Universitaria Senese, 53100 Siena, Italy; 7Department of Radiology, Albert Einstein College of Medicine, Montefiore Medical Center, Bronx, NY 10467-2490, USA; 8Department of Interventional and Diagnostic Neuroradiology, Neurocenter of Southern Switzerland, EOC, 6900 Lugano, Switzerland; 9Beam Interventional & Diagnostic Imaging, Department of Interventional Pain Management, Calgary, AB 2500, Canada; 10Department of Radiology, University of California, San Diego, CA 92025, USA; 11Department of Orthopaedic Sugery and Traumatology, HFR Hôpital Cantonal, Unibversity of Fribourg, CH-1708 Fribourg, Switzerland; 12Minimal Invasive Spine Department of Neurosurgery, Istituto Oncologico del Mediterraneo IOM, 95029 Viagrande, Italy; 13UOC Neuroradiologia AO Cardarelli Naples Italy, 80131 Napoli, Italy; 14Department of Radiology, Jacobi Medical Center/Albert Einstein College of Medicine, Bronx, NY 11501, USA; 15Department of Radiology, Azienda Ospedaliero Universitaria (A.O.U.) di Cagliari, 09100 Cagliari, Italy; 162nd Department of Radiology, University General Hospital “ATTIKON” Medical School, National and Kapodistrian University of Athens, 12462 Athens, Greece; 17UOC Radiologia SS, Trinità Hospital, 09121 Cagliari, Italy; 18Diagnostica per Immagini e Radiologia Interventistica, Università di Roma Tor Vergata, 00148 Roma, Italy

**Keywords:** pain, vertebroplasty, ablation, augmentation, kyphosis, target, future, treatment

## Abstract

Vertebral augmentation has been used to treat painful vertebral compression fractures and metastatic lesions in millions of patients around the world. An international group of subject matter experts have considered the evidence, including but not limited to mortality. These considerations led them to ask whether it is appropriate to allow the subjective measure of pain to so dominate the clinical decision of whether to proceed with augmentation. The discussions that ensued are related below.

In her classic song, *Haven’t got time for the pain*, Carly Simon famously finished by stating, “the time for the pain is over.” As it relates to vertebral compression fractures (VCFs), the multidisciplinary group of subject matter experts serving as faculty at the Sardinian Spine and Stroke 2022 conference propose modifying Ms. Simon’s tagline in order to rethink the perseveration on pain palliation that has typified the approach towards VCFs [1]. Vertebral augmentation has changed the landscape for patients suffering from painful vertebral compression fractures [2]. What had previously been a disease lacking effective treatments beyond non-surgical management (NSM) could now reliably, and with low complication rates, get patients out of the hospital, with rapid mobilization and a return to normal function [3,4].

Radiologists were critical of the emergence of this then-nascent approach to treating VCFs [5]. For many radiologists, this may have been their first exposure to treating pain. Perhaps understandably, an orthodoxy began to develop and held the viewpoint that absent significant pain, vertebral augmentation should not be performed. There is a clear logic to this, as pain treatments should never be performed simply to address imaging findings. However, it might be possible that this approach is hindering advancement in the field and, more importantly, limiting appropriate access to the procedure. Pain assessments are inherently subjective, and the literature abounds in inconsistencies with respect to patient assessment, validated pain scales, patient follow-up methodologies, confounding sham treatments and placebo effects [6,7]. Re-focusing attention on the biomechanical effect of percutaneous augmentation allows one to utilize an objective platform to assess these interventions with reproducible, measurable outcomes. One can then show the efficacy of these procedures in terms of their impact on the functional spine unit, including the disc–endplate complex and facet joints, and on sagittal balance, with off-loading of the anterior vertebral column secondary to height restoration and prevention of further height loss of treated vertebrae.

The blinded trials that have followed patients for extended periods after randomization have convincingly demonstrated a preservation of sagittal height relative to sham patients from the same trials [8,9]. Traditionally, the preservation of vertebral height alone would not generally be considered by many practitioners significant enough to warrant the risk of intervention [10]. However, the faculty of the Sardinian conference wondered if this rationale is entirely appropriate. Anterior column fractures will, over time, often lead to hyperkyphosis, disc degeneration, facet osteoarthritis, spinal canal stenosis and other causes of chronic pain [11,12,13]. Early intervention or augmentation when the fracture is still able to be optimally reduced might stave off these undesirable secondary effects, such as reducing adjacent-level creep deformation [14]. Every patient experiences an informed consent prior to a procedure where the risks are detailed, but none of the faculty indicated that they mention the risk of acute or subacute fracture turning into a chronic condition prone to a segmental degenerative change often associated with pain if the vertebra is not treated.

Demonstrating the potentially misleading nature of retrospective case series, relatively early in the U.S. vertebral augmentation experience Uppin et al. concluded that augmentation was associated with an increased incidence of additional fractures [15]. This highly cited paper soon became a focal point for those who did not support/believe in augmentation despite the fact that when compared with non-surgical management, patients did much better with augmentation [16,17]. A few years later, various meta-analyses comparing augmentation to either conservative therapy or sham have demonstrated that adjacent level fractures following augmentation matches natural history [18,19,20]. While that literature is substantial in disproving the Uppin case series, the more recent SAKOS trial and other data suggests that subtypes of augmentation might have lower levels of adjacent level fractures than traditional vertebroplasty or kyphoplasty [21,22]. This would then imply that instrumented augmentation and implants other than polymethylmethacrylate might decrease adjacent level fracture rates to levels below that of the natural history of the disease. While there is no consensus that these studies are authoritative in making this aspirational point, if true, the rationale for an augmentation becomes ever stronger. Put differently, should a patient who has relatively modest pain be denied the potential benefit of lowering the natural history of adjacent level fractures? Should that patient with modest pain be denied the potential benefit of a vertebra with at least partially restored morphology, ability to sustain axial load, and ultimately more physiological biomechanical function? Beyond that, questions can be raised on the role of prophylactic vertebroplasty though that is beyond the intended scope of this comment [23].

Cianfoni et al. have published multiple studies looking at the stent and screw-assisted internal fixation (SAIF) technique [24,25,26,27,28]. Utilizing this method, extremely complex osteoporotic and neoplastic fractures can be approached, stabilized, and anatomically corrected, confirming results from preliminary biomechanical studies. The alternative to SAIF in many of these vulnerable patients would be reconstructive open spine surgery. Cianfoni, as faculty of the Sardinian conference, noted that a common theme for these SAIF paper submissions is that reviewers typically insist the authors provide Visual Analog Score or Numeric Rating Score data. When one considers the numerous publications supporting pain relief utilizing traditional augmentation including multiple randomized control trials, it is reasonable to wonder what these small but extreme patient series would contribute to the understanding of how cementation impacts mechanical pain from fractures [29]. For the subgroup of these extreme SAIF patients, more relevant measures include how frequently patients avoided conventional surgery, the ability to obtain vertebral body reconstruction, correction of posterior wall encroachment, the degree of improvement in sagittal balance, and how safe and durable these procedures’ biomechanical advantages are [30,31]. Nonetheless, the treatment of pain is so ingrained in the minds of augmentation practitioners, it becomes a dominant question for expert reviewers seemingly every time one of these papers is subjected to peer review. We claim that vertebral augmentation (VA) is superior to Non-Surgical Management (NSM). Additionally, concerning more complex fractures, where posterior tension band is injured and/or screws are needed, we claim that VA+Vertebral Posterior Fixation (VPF) is superior to classical gold standard stand-alone VPF. Even in many cases in which a double approach, i.e., anterior and posterior surgical approach of the spine, is performed, vertebral augmentation can play a role in avoiding the anterior approach and minimize comorbidities from the procedure, with better short-term clinical results, and similar clinical and radiological medium- and long-time results.

OPuS One was a landmark study looking at radiofrequency ablation (RFA) of spinal metastases [32]. Due to ethical considerations and provider/patient preference, trial participants could be treated with RFA alone, but the vast majority of patients underwent both RFA and cementation. This study is appropriately reported as very supportive of the techniques and has likely contributed to greater utilization of RFA in these types of patients. Indeed, the consensus opinion of the authors of this comment is that the increasing use of RFA in this cohort is good for the patients undergoing these procedures. Once again, however, the subject matter experts of the Sardinian conference are left to wonder if pain was the optimized measurement for this cohort. The authors of this report would argue that it was not. When one compares the results of the CAFÉ trial to OPuS One (i.e., cementation with and without RFA) it is difficult to conclude that pain control was improved by RFA [33,34]. When one considers the rationales for an ablative technique including procedural safety, local tumor control, debulking of disease or decreasing the potential for tumor spread, one can wonder about the value of studying pain as the primary endpoint [35,36]. Indeed, the potential benefit of these interventions in cancer patients with spine involvement, although previously understated, can be attributed to the prevention of spinal instability and spinal canal compromise. These are objective measurable endpoints which can be assessed at follow-up and have important biomechanical implications. A spinal instability score is obtained at the time of the patient’s clinical presentation; it seems intuitive that the score can be improved following these percutaneous interventions [37,38].

It is evident that vertebral fractures contribute to mortality and lower survival of patients; furthermore, hospitalization burden and cost of VCFs management has been shown to be higher than that of myocardial infarct, cerebrovascular accident or breast cancer [39,40]. The accepted threshold for complications related to vertebral augmentation techniques is in the range 2.2–3.9% for osteoporotic fractures [4]. The literature data clearly suggest that complications from performing vertebral augmentation are less than complications from not performing the procedure. A growing body of literature demonstrates improved mortality in patients undergoing vertebral augmentation as compared with patients treated with conservative therapy [41,42,43]. Ong et al. found a statistically significant difference comparing time matched periods before and after the equivocal 2009 NEJM studies, which the authors hypothesized might relate to a decrease in utilization of vertebral augmentation procedures [44,45]. This investigation was followed by a meta-analysis of sixteen studies that convincingly made this point [46]. Utilizing the same claims-based dataset, Ong, et al. then converted the survival probability data from this 10-year period into a number needed to treat analysis and found remarkably small numbers of patients that would need to be treated to lead to a difference in survivability [47]. The authors also speculated on the reasons for the improved mortality with rationales ranging from decreased opioid requirements, greater mobility, and sagittal balance preservation with Oxford Level 2a evidence supporting these mortality claims [46]. The rationale for the improved survival in patients treated with vertebral augmentation, while not completely certain, is likely multifactorial. What if sagittal balance preservation or improvement is a meaningful part of that claim? In a situation where augmentation provides a mortality benefit, is it appropriate to limit treatment to those with significant pain? [48]. Future research upon vertebral augmentation techniques could utilize more deliberate and objective methods with more extensive evaluation factors and the inclusion of more patients with different baselines who will be followed up for longer periods of time [49].

The three days in Cagliari raised many questions around a common theme. Augmentation has been reported in thousands of publications to be an effective treatment for painful vertebral compression fractures. In that clinical scenario, the literature largely believes it to be successful [50,51,52]. Emerging arguments support the use of vertebral augmentation for sagittal balance preservation and restoration. Perhaps that preservation plays an important role in the strong mortality benefit associated with augmentation. Cancer patients undergoing radiofrequency ablation are not necessarily evaluated in the way other ablative, e.g., radiation therapy occurs but rather as it relates to pain relief. Cianfoni describes techniques that resolve the issue of undertreatment of the anterior column alone via augmentation with multicolumn support vs. overtreatment in these vulnerable patients. While patients treated with SAIF would typically be treated surgically where pain relief would be a secondary rationale, reviews of these submissions often focus on that as a primary component. Thirty-five years after the first vertebroplasty, it may be time to target something other than pain [53]. The knowledge accumulated during this period suggests that we should focus on some other aspects of vertebral compression fractures: prevention of progressive collapse, chronic pain, adjacent level fractures; anatomical restoration of the vertebrae and sagittal balance restoration; correction of kyphotic deformity. The author’s consensus is that broadening our treatment focus will improve patients’ clinical results. Perhaps, to play on Carly Simon’s famous lyrics, in some patients, we have reached a point where the “need for the pain” palliation should be considered along with the other demonstrable benefits of vertebral augmentation.

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
