# Peer review of "Vertebral Augmentation: Is It Time to Get Past the Pain? A Consensus Statement from the Sardinia Spine and Stroke Congress"

_medicina, 2022, doi:10.3390/medicina58101431_

Round 1
Reviewer 1 Report
The paper under review is preneted as a Statement from the Sardinia Spine and Stroke Congress concerning the role of Vertebral Body Augmentation in current practice. However, I had serious issue in reviewing this document. First and foremost, I cannot understand the scope of this paper: is this a Review? If yes, narrative, systematic or what else? If, as the Author's claim, is a Consenusus statement, the Authors should describe the process used to reach the so called consensus (eg Delphi Consensus...); finally, what is the consensus? what are the questions that this paper tries to answer?
Because as for now, the paper reads as a divulgative paper, not a scientific paper.
Therefore, it is my honest opinion that the paper is not accetable in its current form.
Author Response
Response to Reviewer 1 Comments
Point 1: The paper under review is preneted as a Statement from the Sardinia Spine and Stroke Congress concerning the role of Vertebral Body Augmentation in current practice. However, I had serious issue in reviewing this document. First and foremost, I cannot understand the scope of this paper: is this a Review? If yes, narrative, systematic or what else? If, as the Author's claim, is a Consenusus statement, the Authors should describe the process used to reach the so called consensus (eg Delphi Consensus...); finally, what is the consensus? what are the questions that this paper tries to answer?
Because as for now, the paper reads as a divulgative paper, not a scientific paper.
Therefore, it is my honest opinion that the paper is not accetable in its current form.
Response 1: We thank reviewer one for his/her comments. While the English language authors were not completely familiar with the term “divulgative, we believe the points he/her were trying to make are clear, i.e., he/she does not understand the “scope of the paper.” As described, the authors were all part of an international group of subject matter experts/invited faculty at the semi-annual Sardinia Spine meeting. A consensus opinion emerged over the course of several days. The corresponding authors believed this would be a terrific piece to present to Medicina and thus the paper was submitted. The paper is thus exactly what it purports to be, a consensus statement of subject matter experts. No process, e.g., Delphi was utilized as that was not the intent. The authors were simply making a comment based on their consensus opinion.
Reviewer 2 Report
Group of authors - orhopedic surgeons, radiologist, neuroradiology and neurosurgery - present a consensus statement on vertebral augmentation following Sardinia Spine and Stroke congress. Manuscript is more written as an experts opinion at one congress, then as a consensus statement. Otherwise, it needs English language editing - for example, authors write on several occasions improved mortality and improved survival - I suggest to use uniform language.
For a consensus statement, this manuscript lacks several point : time frame in which current literature was analyzed with methods, i.e. inclusion and exclusion parameters; furthermore, consensus reports usually contain several conclusions on which the congress members have voted. Consensus statement needs to be clearly made in several point-to-point sentences.
One important question which needs to be clarified - do the authors consider vertebral augmentation superior to non-surgical treatment OR superior to other surgical modalities, such as stabilization? There are also situations where stabilization and augmentation need to be performed simultaneously. Augmentation - kyphoplasty, elastoplasty - has certainly a role in spine surgery, however in complex instable fractures it cannot replace stabilization surgery alone.
I suggest to include several references on bias on reporting vertebral augmentation and on the subject, comment and discuss:
Beall DP, Tutton SM, Murphy K, Olan W, Warner C, Test JB. Analysis of Reporting Bias in Vertebral Augmentation. Pain Physician. 2017 Nov;20(7):E1081-E1090. PMID: 29149153.
What are your counter-arguments for recommendations issued from endocrinologists and rheumatologists, who do not recommend routine use of the method:
Ebeling PR, Akesson K, Bauer DC, Buchbinder R, Eastell R, Fink HA, Giangregorio L, Guanabens N, Kado D, Kallmes D, Katzman W, Rodriguez A, Wermers R, Wilson HA, Bouxsein ML. The Efficacy and Safety of Vertebral Augmentation: A Second ASBMR Task Force Report. J Bone Miner Res. 2019 Jan;34(1):3-21. doi: 10.1002/jbmr.3653. PMID: 30677181.
Include and comment:
Cornelis FH, Joly Q, Nouri-Neuville M, Ben-Ammar M, Kastler B, Kastler A, Amoretti N, Hauger O. Innovative Spine Implants for Improved Augmentation and Stability in Neoplastic Vertebral Compression Fracture. Medicina (Kaunas). 2019 Jul 31;55(8):426. doi: 10.3390/medicina55080426. PMID: 31370309; PMCID: PMC6722751.
There is a need to include risk-benefit evaluation of the complications of the procedure too:
Gao X, Du J, Gao L, Hao D, Hui H, He B, Yan L. Risk factors for bone cement displacement after percutaneous vertebral augmentation for osteoporotic vertebral compression fractures. Front Surg. 2022 Jul 28;9:947212. doi: 10.3389/fsurg.2022.947212. PMID: 35965863; PMCID: PMC9366098.
Wang S, Zheng L, Ma JX, Wang H, Sun ST, Zhang BH, Guo XL, Xiang LB, Chen Y. Analysis of the most influential publications on vertebral augmentation for treating osteoporotic vertebral compression fracture: A review. Medicine (Baltimore). 2022 Aug 5;101(31):e30023. doi: 10.1097/MD.0000000000030023. PMID: 35945791; PMCID: PMC9351837.
Author Response
Response to Reviewer 2 Comments
Point 1: Group of authors - orhopedic surgeons, radiologist, neuroradiology and neurosurgery - present a consensus statement on vertebral augmentation following Sardinia Spine and Stroke congress. Manuscript is more written as an experts opinion at one congress, then as a consensus statement. Otherwise, it needs English language editing - for example, authors write on several occasions improved mortality and improved survival - I suggest to use uniform language.
Response 1: We thank reviewer 2 for this comment. The manuscript includes multiple native English language speaker authors who have re-reviewed and stand by the manuscript as it is written.
Point 2: For a consensus statement, this manuscript lacks several point : time frame in which current literature was analyzed with methods, i.e. inclusion and exclusion parameters; furthermore, consensus reports usually contain several conclusions on which the congress members have voted. Consensus statement needs to be clearly made in several point-to-point sentences.
Response 2: We thank reviewer 2 for this comment. As in our response to Reviewer 1, this was simply the completely agreed upon consensus of all authors. There was no voting or other formal method. A specific author took notes and constructed the initial draft. One by one (literally), every author was asked and then provided their opinions and feedback. The points made in this paper are supported by all of the authors
Point 3: One important question which needs to be clarified - do the authors consider vertebral augmentation superior to non-surgical treatment OR superior to other surgical modalities, such as stabilization? There are also situations where stabilization and augmentation need to be performed simultaneously. Augmentation - kyphoplasty, elastoplasty - has certainly a role in spine surgery, however in complex instable fractures it cannot replace stabilization surgery alone.
Response 3: We thank reviewer 2 for this comment. While we agree with the reviewer, his/her point is clearly beyond the scope of this consensus statement which in no way addresses the questions being posed. While we could make a statement along the lines of what is suggested, we think it would be out of place and potentially confusing. We ask the editorial staff to weigh in. Our advice would be to hold off making such a statement. We claim that vertebral augmentation (VA) is superior to Non Surgical Management (NSM). And concerning more complex fractures, where posterior tension band is injured and / or screws are needed, we claim that VA+Vertebral Posterior Fixation (VPF) is superior to classical gold-standard VPF stand alone. Even that in many cases in which double approach, anterior and posterior surgical approach of the spine, is performed, vertebral augmentation can play a role to avoid the anterior approach and minimize comorbidities from the procedure, with better short-term clinical results, and similar clinical and radiological medium-and long-time results.
Point 4: I suggest to include several references on bias on reporting vertebral augmentation and on the subject, comment and discuss:
Beall DP, Tutton SM, Murphy K, Olan W, Warner C, Test JB. Analysis of Reporting Bias in Vertebral Augmentation. Pain Physician. 2017 Nov;20(7):E1081-E1090. PMID: 29149153.
Response 4: We thank reviewer 2 for this comment. Dr. Beall is one of the subject matter expert authors and while we would be delighted to include this paper as a reference, it is really not on point with our consensus comment. We ask the editorial staff to weigh in. Our advice would be to hold off including this paper as a reference. If the editorial staff disagree, please let us know how they would like us to include.
Point 5: What are your counter-arguments for recommendations issued from endocrinologists and rheumatologists, who do not recommend routine use of the method:
Ebeling PR, Akesson K, Bauer DC, Buchbinder R, Eastell R, Fink HA, Giangregorio L, Guanabens N, Kado D, Kallmes D, Katzman W, Rodriguez A, Wermers R, Wilson HA, Bouxsein ML. The Efficacy and Safety of Vertebral Augmentation: A Second ASBMR Task Force Report. J Bone Miner Res. 2019 Jan;34(1):3-21. doi: 10.1002/jbmr.3653. PMID: 30677181.
Response 5: We thank reviewer 2 for this comment. The authors are all familiar with this Eberling et al paper. We ask the editorial staff to weigh in as our subject matter expert comment is that of proponents of the procedure who believe augmentation is being underutilized in a variety of situations. Eberling et al in this ASBMR Task force report disagree. In the same way Eberling et al would not consider opinion of proponents of the procedure in any consensus opinion they would create, it seems out of place in this comment. Our advice would be to hold off on including this material.
Point 6: Include and comment: Cornelis FH, Joly Q, Nouri-Neuville M, Ben-Ammar M, Kastler B, Kastler A, Amoretti N, Hauger O. Innovative Spine Implants for Improved Augmentation and Stability in Neoplastic Vertebral Compression Fracture. Medicina (Kaunas). 2019 Jul 31;55(8):426. doi: 10.3390/medicina55080426. PMID: 31370309; PMCID: PMC6722751.
Response 6: We thank reviewer 2 for this comment. We would be delighted to include this reference in the section on SAIF. No additional language would be required.
Point 7: There is a need to include risk-benefit evaluation of the complications of the procedure too:
Gao X, Du J, Gao L, Hao D, Hui H, He B, Yan L. Risk factors for bone cement displacement after percutaneous vertebral augmentation for osteoporotic vertebral compression fractures. Front Surg. 2022 Jul 28;9:947212. doi: 10.3389/fsurg.2022.947212. PMID: 35965863; PMCID: PMC9366098.
Wang S, Zheng L, Ma JX, Wang H, Sun ST, Zhang BH, Guo XL, Xiang LB, Chen Y. Analysis of the most influential publications on vertebral augmentation for treating osteoporotic vertebral compression fracture: A review. Medicine (Baltimore). 2022 Aug 5;101(31):e30023. doi: 10.1097/MD.0000000000030023. PMID: 35945791; PMCID: PMC9351837
Response 7: We thank reviewer 2 for this comment. We are not sure what role either of these papers would have with respect to the consensus opinion being provided and have thus not included. If the editorial staff recommends that we add as references, we will go ahead and do so.
Reviewer 3 Report
This commentary paper is a great effort to expand current indications of vertebral augmentation techniques.
Authors are advised to provide comments upon application of vertebral augmentation in relation to pathologic fractures post radiation therapy and expand a bit the comment upon local tumor control where ablation is combined with vertebral augmentation in all cases
Author Response
Response to Reviewer 3 Comments
Point 1: This commentary paper is a great effort to expand current indications of vertebral augmentation techniques.
Authors are advised to provide comments upon application of vertebral augmentation in relation to pathologic fractures post radiation therapy and expand a bit the comment upon local tumor control where ablation is combined with vertebral augmentation in all cases
Response 1: We thank reviewer 3 for this comment. While this reviewer is obviously supportive, our point was a bit different in that we weren’t making comments re: supporting the use of vertebral augmentation in different pathologic situations. Rather, we were making a point regarding the challenges inherent in thinking about pain control as the absolutely dominant indication for augmentation. For that reason, we would recommend against inclusion of what reviewer 3 requests
Reviewer 4 Report
This communication is very relevant to not only physicians/medical staff performing vertebral augmentation procedures, but also to those responsible for allowing access to these procedures (i.e. insurance companies). As mentioned in this communication, pain is subjective and not always present in cases in which augmentation would be beneficial to prevent morbidity and mortality. I am in agreement that objective criteria irrespective of the presence of pain should also be considered.
Author Response
Response to Reviewer 4 Comments
Point 1: This communication is very relevant to not only physicians/medical staff performing vertebral augmentation procedures, but also to those responsible for allowing access to these procedures (i.e. insurance companies). As mentioned in this communication, pain is subjective and not always present in cases in which augmentation would be beneficial to prevent morbidity and mortality. I am in agreement that objective criteria irrespective of the presence of pain should also be considered.
Response 1: We thank reviewer 4 for this comment.
Round 2
Reviewer 1 Report
The paper was not changed. I remain of the opinion that as it is, the paper is not suitable for publication.
Author Response
Response to Reviewer 2 Comments
The authors provided a detailed answer on reviewers remarks, however almost none of the reviewers suggestions was included in the final draft.
Point 1: I suggest to include Response 3. into the body of the manuscript, since stabilization vs. augmentation question is very important in surgical management of spinal fractures.
Response 1: we thank the reviewer for his/her comment; added to the text : “We claim that vertebral augmentation (VA) is superior to Non Surgical Management (NSM). And concerning more complex fractures, where posterior tension band is injured and / or screws are needed, we claim that VA+Vertebral Posterior Fixation (VPF) is superior to classical gold-standard VPF stand alone. Even that in many cases in which double approach, anterior and posterior surgical approach of the spine, is performed, vertebral augmentation can play a role to avoid the anterior approach and minimize comorbidities from the procedure, with better short-term clinical results, and similar clinical and radiological medium-and long-time results.”
Point 2: Response 5. (What are your counter-arguments for recommendations issued from endocrinologists and rheumatologists, who do not recommend routine use of the method: beling PR, Akesson K, Bauer DC, Buchbinder R, Eastell R, Fink HA, Giangregorio L, Guanabens N, Kado D, Kallmes D, Katzman W, Rodriguez A, Wermers R, Wilson HA, Bouxsein ML. The Efficacy and Safety of Vertebral Augmentation: A Second ASBMR Task Force Report. J Bone Miner Res. 2019 Jan;34(1):3-21. doi: 10.1002/jbmr.3653. PMID: 30677181.) is just not sufficient - I agreee with authors and disagree with viewpoint of Eberling et al.; however, a certain explanation and counter-arguments would in my opinion fortify the consensus statement.
Response 2: we thank the reviewer for his/her comment; added to the text: “It is evident that vertebral fractures contribute to mortality and lower survival of patients; furthermore, hospitalization burden and cost of VCFs management has been shown to be higher than that of myocardial infarct, cerebrovascular accident or breast cancer [39, 40]. Accepted threshold for complications related to vertebral augmentation techniques range from 2.2-3.9% for osteoporotic fractures [4]. Literature data clearly suggest that complications from performing vertebral augmentation are less than complications from not performing the procedure.”
Point 3: As for Point 7. of the previous revision (There is a need to include risk-benefit evaluation of the complications of the procedure too: Gao X, Du J, Gao L, Hao D, Hui H, He B, Yan L. Risk factors for bone cement displacement after percutaneous vertebral augmentation for osteoporotic vertebral compression fractures. Front Surg. 2022 Jul 28;9:947212. doi: 10.3389/fsurg.2022.947212. PMID: 35965863; PMCID: PMC9366098.Wang S, Zheng L, Ma JX, Wang H, Sun ST, Zhang BH, Guo XL, Xiang LB, Chen Y. Analysis of the most influential publications on vertebral augmentation for treating osteoporotic vertebral compression fracture: A review. Medicine (Baltimore). 2022 Aug 5;101(31):e30023. doi: 10.1097/MD.0000000000030023. PMID: 35945791; PMCID: PMC9351837) - my opinion is, if you make a consensus statement on any kind of procedure, including complications of these procedures into the final draft is of utmost importance.
Response 3: we thank the reviewer for his/her comment; added to the text: “It is evident that vertebral fractures contribute to mortality and lower survival of patients; furthermore, hospitalization burden and cost of VCFs management has been shown to be higher than that of myocardial infarct, cerebrovascular accident or breast cancer [39, 40]. Accepted threshold for complications related to vertebral augmentation techniques range from 2.2-3.9% for osteoporotic fractures [4]. Literature data clearly suggest that complications from performing vertebral augmentation are less than complications from not performing the procedure.”
Also added to the text: “Future research upon vertebral augmentation techniques could utilize more deliberate and objective methods with more extensive evaluation factors and inclusion of more patients with different baselines who will be followed-up for longer period of time [49]”
Reviewer 2 Report
The authors provided a detailed answer on reviewers remarks, however almost none of the reviewers suggestions was included in the final draft. I suggest to include Response 3. into the body of the manuscript, since stabilization vs. augmentation question is very important in surgical management of spinal fractures. Response 5. is just not sufficient - I agreee with authors and disagree with viewpoint of Eberling et al.; however, a certain explanation and counter-arguments would in my opinion fortify the consensus statement. As for Point 7. of the previous revision - my opinion is, if you make a consensus statement on any kind of procedure, including complications of these procedures into the final draft is of utmost importance.
Author Response

(The authors gave the same response as above.)
